# Traumatic brain injuries and problem gambling in youth: Evidence from a population-based study of secondary students in Ontario, Canada

Nigel E. Turner [1,2,3‡*], Steven Cook[4‡], Jing Shi[1,5°], Tara Elton-Marshall[1,2,3°], Hayley Hamilton[1,2,3°], Gabriela Ilie[6°], Christine M. Wickens[1,2,3,7°], André J. McDonald[1,2°], Nico Trajtenberg[8°], Michael D. Cusimano[2,9°], Robert E. Mann[1,2,3°]

1 Institute for Mental Health Policy Research, Centre for Addiction and Mental Health, Toronto, Ontario, Canada, 2 Dalla Lana School of Public Health, University of Toronto, Toronto, Ontario, Canada, 3 Campbell Family Mental Health Research Institute, Centre for Addiction and Mental Health, Toronto, Ontario, Canada, 4 Department of Epidemiology, University of Michigan School of Public Health, Ann Arbor, Michigan, United States of America, 5 School of Rehabilitation Science, McMaster University, Hamilton, Ontario, Canada, 6 Faculty of Medicine, Dalhousie University, Halifax, Canada, 7 Institute of Health Policy, Management and Evaluation, University of Toronto, Toronto, Ontario, Canada, 8 School of Social Sciences, Cardiff University, Cardiff, Wales, United Kingdom, 9 Division of Neurosurgery, Department of Surgery, St. Michael's Hospital, Toronto, Ontario, Canada

☯ These authors contributed equally to this work.
‡ NET and SC are joint senior authors on this work.
* Nigel.Turner@camh.ca

**Data Availability Statement:** Due to institutional restrictions data cannot be made available in the manuscript, the supplemental files or a public

## Abstract

Traumatic brain injury (TBI) is characterized by a change in brain function after an external force or sudden movement to the head. TBI is associated with risk-taking, impulsivity, psychological distress, substance abuse, and violent crime. Previous studies have also linked problem gambling to TBI, but these studies have not controlled for possible confounding variables such as mental health problems and hazardous drinking which are also linked to TBI. This study examines the relationship between problem gambling and TBI among adolescents. Data were obtained from the 2011, 2013 and 2015 cycles of the OSDUHS, a biennial cross-sectional school-based study of children in grades 7 to 12 ($N = 9,198$). Logistic regression was used to estimate adjusted odds ratios (AOR) in controlled and uncontrolled analyses. Adjusting for sex and grade only, problem gambling was associated with a history of TBI (AOR = 2.8). This association remained significant after adjusting for hazardous drinking and suicidality (AOR = 2.0). In addition, problem gambling had a statistically significant relationship with being male (AOR = 4.7), hazardous drinking (AOR = 4.5), and suicidality (AOR = 3.1). This study provides further data to suggest a link between TBI and problem gambling. However, research is needed on the causal relationship between these variables and the potential implications for treatment and prevention.

repository. Data are available to all researchers after an institutional application is approved. To access data contact osduhs@camh.ca for further instructions. Full reports of the public data files and the surveys administered can be accessed online: www.camh.ca/osduhs.

**Funding:** This research is based on the Ontario Student Drug Use and Health Survey, a Centre for Addiction and Mental Health initiative funded in part through ongoing support from the Ontario Ministry of Health and Long-Term Care, as well as targeted funding from several provincial agencies. Additional funding was provided by the Ministry of Health and Long Term Care as grant number 06703. The funders had no role in study design, data collection and analysis, decision to publish, or preparation of the manuscript.

**Competing interests:** Turner has received funding from the Ontario Ministry of Health and Long Term Care (OMHLTC), and Gambling Research Exchange (GREO). Turner has also acted as a consultant on gambling problems for various government and legal entities. For another project unrelated to this one Turner, has received grant funding from the Ontario Lottery and Gaming (the crown corporation that manages gambling in Ontario) to evaluate some of their prevention initiatives, but otherwise has not received funding from the gambling industry. The contract included guarantees of independence and intellectual property rights for the researcher. The remaining authors declare no competing interests. None of these conflicts or sources of funding alter our adherence to PLOS ONE policies on sharing data and materials.

**Abbreviations:** GD, Gambling Disorder; OSDUHS, Ontario Student Drug Use and Health Survey; CAMH, Centre for Addiction and Mental Health; AUDIT, Alcohol Use Disorders Identification Test; AOR, Adjusted odds ratios; TBI, Traumatic brain injury; SOGS-RA, South Oaks Gambling Screen Revised for Adolescents; CAGI, Canadian Adolescent Gambling Index.

## Introduction

Studies of gambling among adolescents have found that despite legal restrictions many adolescents report participating in gambling [1–4]; most often private bets and card games [5]. Problem gambling among adolescents can seriously affect adolescents' social, family, academic, and financial lives A recent study [6] reported that 1% of secondary students fell into the "red" category on the Canadian Adolescent Gambling Index (CAGI) indicating that they scored in the range of a severe gambling problem and an additional 3.3% of students scored in the "yellow" category indicating that they had a subclinical level of symptoms. In addition, research has shown that most male severe problem gamblers report starting gambling during adolescence [7–9].

A variety of social and individual risk factors have been found to be associated with problem gambling including being male, using alcohol and drugs, anxiety, depression, poor grades, delinquency, poor decision making, antagonism, disinhibition, and suicidal thoughts [1, 2, 10–16]. Additional understanding of the factors that put people at risk for a gambling problem is necessary for preventing and treating this disorder. Another potential risk factor that may be linked to problem gambling is Traumatic Brain Injury (TBI) [17].

Traumatic Brain Injury (TBI) is caused by a blow to the head by an external force, or sudden movement, that can lead to serious long-term consequences [18–20]. TBI is a common form of injury that can result in changes in behavior, emotion and/or cognitive function [27–29]. Individuals reporting past TBI are also more likely to report substance misuse, aggressive behaviors, and mental health and substance use problems [21–24]. Given the relationships of TBI with alcohol consumption, other substance use, and mental health problems [21, 23, 25], it seems plausible that adolescent TBI may also be related to problem gambling. An emerging literature with adults supports this view. Whiting et al. [26], for example, observed that in a sample of 738 American veterans, TBI was significantly correlated with problem gambling. In a community derived sample of 104 problem gamblers, Hodgins and Hollub [27] found that nearly half reported experiencing a TBI in their lifetime. Similarly, Turner et al. [17] found that people who reported experiencing one or more TBIs in their lifetime had more than 3 times the odds of also reporting having problems with gambling [17]. The link between TBI and problem gambling remained statistically tenable after controlling for hazardous drinking and psychological distress [17]. Similarly, a recent case-control study by Bhatti et al. [28] found that prior TBI was a significant predictor of subsequent gambling problems, particularly among middle-aged men, those who reported alcohol or tobacco use, and those who experienced multiple TBIs. The link between TBI and gambling is also supported by recent neurological research [29] and animal research [30].

However, we did not find any study that examined the association between problem gambling and TBI in an adolescent sample. This is surprising given the evidence suggesting that TBIs are common among adolescents [31], that problem gambling may have origins in the adolescent years [7, 9], and that TBI and gambling share a number of comorbidities (e.g., substance abuse, depression). Interestingly we did find reports of poor performance on gambling-based decision making tasks in adults [29, 30] and in youth [32], and gambling is mentioned as a potential barrier for the treatment of TBI [33]. Studies of TBI have shown that it is associated with impulsiveness and a variety of negative mental health outcomes including depression, anxiety, suicidal ideation and substance abuse [22, 25, 34], many of which are also comorbid with problem gambling [13–15, 35].

The current study examined the association between TBI and problem gambling in a population-based sample of secondary school students from Ontario, Canada. To our knowledge, this study is the first to examine the association between problem gambling, hazardous

drinking, suicidal ideation, and TBI in a population-based study of youth. Because of the comorbidity between problem gambling, hazardous drinking and suicidal ideation [14] on the one hand, and TBI, hazardous drinking and suicidal ideation [22, 25] on the other hand, we conducted an analysis of TBI as a predictor of gambling problems controlling for hazardous drinking and suicidal ideation. This allowed us to determine if there is a direct link between TBI and gambling or if it is merely a result of overlapping comorbidities.

## Materials and methods

### Sample

The data for the current study were derived from a subsample of secondary school (grades 9–12) students from the 2011, 2013 and 2015 cycles of the Ontario Student Drug Use and Health Survey (OSDUHS), a repeated cross-sectional population survey of students from Ontario, Canada. Given the relative infrequency of problem gambling among secondary school students, we leveraged the repeated structure of the OSDUHS and merged these three waves of survey data to increase statistical power. These waves were selected because they contained the same questions for demographics, gambling problems, TBI, hazardous drinking, and suicidality. At each survey wave, a random half-sample of participating students were asked questions about their head injuries and gambling problem behaviors. The total sample size was 9,198 secondary school students (52.0% female), with a mean age of 15.8 years (range: 11–20; SD = 1.26), and 98.9% between the ages of 14–18. Each wave of the survey received ethics approval from the Research Ethics Boards of York University and the Centre for Addiction and Mental Health (Protocols:#062/2010 for OSDUHS 2011,: #068/2012 for OSDUHS 2013, and #018/2014 for OSDUHS 2015). All procedures performed in studies involving human participants were in accordance with the ethical standards of the institutional and/or national research committee and with the 1964 Helsinki Declaration and its later amendments including informed consent and confidentiality of all personal information. An active consent method was used. A description of the study and a consent form was sent home with the students. To participate in the survey, written parental consent and written student assent were required.

### Measures

Demographic variables included sex (male = 1, female = 0) and grade level (grades 9, 10, 11, 12). We control for the effects of grade level and sex in the analysis models.

**Problem gambling.**   The OSDUHS used a shortened version of the South Oaks Gambling Screen Revised for Adolescents (SOGS-RA [36]), referred to as the short SOGS, to assess problem gambling behavior in 2011, 2013, and 2015. The 6-items in the short SOGS were selected to maximize the content and variance of the full SOGS-RA with a minimum number of items [37]. Receiver operating characteristics determined that a cut-off score of 2 on the short SOGS corresponded with a cut-off of 4 on the full SOGS-RA [14, 37]. In addition, a recent study by Turner found that the cut off of 2 on the Short SOGS was equivalent to the red or severe category on the CAGI [6]. In the current study, a score of two or more on the short SOGS was therefore used to indicate the presence of problem gambling (0 = no, 1 = yes).

**Traumatic brain injury (TBI).**   TBI was assessed through the following question: "We are interested in any head injury that resulted in you being unconscious (knocked out) for at least 5 minutes, or you had to stay in the hospital for at least 1 night because of it. Did you have this type of head injury in your life?" Responses were recoded to create a binary lifetime TBI measure (0 = no, 1 = yes). This definition of lifetime TBI is consistent with the diagnostic criteria outlined in the Diagnostic and Statistical Manual of Mental Disorders, 5th edition [38] and has

been used previously in other research studies [20, 39, 40]. This is a preliminary measure at the population level, and excludes milder injuries when the person lost consciousness for less than 5 minutes.

**Hazardous drinking.** The OSDUHS included the Alcohol Use Disorders Identification Test (AUDIT), which was developed by the World Health Organization [41]. This 10-item instrument is designed to detect hazardous or harmful drinking behavior. Those with a score of 8 or higher out of a maximum of 40 are considered to be drinking at a hazardous or harmful level. The AUDIT has strong internal consistency and test-retest reliability and considered to be a valid measure of alcohol related problems [42]. The AUDIT items focus on the frequency, volume, and pattern of alcohol consumption, as well as adverse consequences associated with drinking and indicators of dependence. Most of the questions are oriented towards the past 12-months. For the current analysis the AUDIT was dichotomized into those scoring 8 or more indicating hazardous or harmful level of drinking and those not reporting such risky behavior (coded 0 = no hazardous drinking, 1 = hazardous drinking).

**Suicidality.** The students were asked 2 questions about suicide behaviors. Specifically, they were asked: "In the past 12 months. . . (1) did you ever seriously consider attempting suicide; (2) did you ever actually attempt suicide?" Response options to both questions were yes or no, and students were considered to have exhibited suicidal thought if they responded yes to either one of these questions (0 = no; 1 = suicide ideation and/or attempt). Both questions are taken from the Centre for Disease Control's Youth Risk Behavior Survey and have been validated among student populations [43]. Note because of a review comment, we re-ran the analysis with suicide attempts as the independent variable as a type of sensitivity analysis. We found very similar results which are reported in supplementary files (see S1 Table) but kept suicide ideation because we believe this to be an important construct in our research.

## Analysis

Our sample of 9,198 respondents was well-distributed among the 31 strata across 320 population sampling units, with subclass observations ranging from 125 to 690 across strata. The OSDUHS employs a stratified (region by school level) two-stage (school, class) cluster design, therefore, variables accounting for the probability of selection, stratification, and clustering are used when analyzing the data. Weights were applied to ensure that the sample distribution reflected that of the Ontario student population [44–46].

All statistical analyses were completed using Stata 16.1 [47]. Weighted means and percentages were calculated for descriptive statistics, and adjusted chi-square tests were used for bivariate analyses. The complex sampling includes some violations of independence of observations due to disproportionate stratification, correlated clusters and diverse inclusion weights, which will result in an underestimate of variances if treated as simple random sampling. To adjust for the complex design we used the SVY command available in STATA which employs weighted (or pseudo) maximum likelihood estimation to compute point estimates and Taylor Series Linearization (a robust nonparametric estimator) to compute variances and confidence intervals [48–50]. Odds ratios (OD) and adjusted odds ratios (AOR) from a series of design-based logistic regression models were used to examine the factors associated with gambling problems among our subpopulation of secondary school students from Ontario. Models controlled for the potential confounding effects of demographic factors, hazardous drinking, and suicidality. Interaction effects were estimated between TBI and hazardous drinking and TBI and suicidality to determine whether the association between TBI and problem gambling was moderated by these variables.

## Results

A total of 1.3% (95% CI: 1.0%-1.7%) of secondary students sampled met the criteria for a gambling problem. Table 1 presents the socio-demographic characteristics for the combined sample and separately based on disordered gambling behavior. Secondary students with problem gambling were more likely to be male (Wald F (1, 298) = 14.47, p<0.001) and report hazardous drinking (Wald F (1, 298) = 54.71, p<0.001). Among problem gamblers, more than half reported hazardous drinking behavior according to their scores on the AUDIT (8 and above). Problem gambling was also associated with suicidal behaviors (Wald F (1, 298) = 18.16, p<0.001), as one third of secondary school problem gamblers reported suicidal ideation and/or attempts during the past twelve months. Traumatic brain injuries were reported by approximately 20% of secondary students without a gambling problem. This was significantly higher for secondary school students with gambling problems, as more than 40% reported a lifetime history of TBI (Wald F (1, 298) = 11.92, p<0.001).

Table 2 presents results from the logistic regression models examining factors associated with problem gambling. All odds ratios are adjusted for sex and grade level. We estimated 3 nested models: Model 1 included TBI (AOR 2.80; 95% CI: 1.43–5.48); Model 2 removed TBI and added hazardous drinking (AOR 5.04; 95% CI:2.79–9.11) and suicidality (AOR 3.27; 95% CI: 1.75–6.11); Model 3 include all of these variables. Although grade was not statistically significant in any of the models, sex was significantly associated with problem gambling in each model (Model 1: AOR = 3.68; Model 2: AOR = 4.94). In Model 3, which adjusted for all variables, the odds of male students having a gambling problem was 4.7 times that of female students (95% CI: 2.18–10.23).

**Table 1. Socio-demographic characteristics of the Ontario secondary students, by gambling problem status, OSDUHS 2011–2015, Ontario Canada, (N = 9,198).**

| | Total Sample | | Non problem (0–1) | | Problem (2+) | | Sig. |
|---|---|---|---|---|---|---|---|
| | % | (95% CI) | % | (95% CI) | % | (95% CI) | |
| **Sex** | | | | | | | *** |
| Female | 52.0 | (49.6–54.4) | 49.4 | (46.8–52.0) | 21.3 | (11.9–35.4) | |
| Male | 48.0 | (45.7–50.4) | 50.6 | (48.0–53.3) | 78.7 | (62.6–88.1) | |
| **Grade** | | | | | | | NS |
| Grade 9 | 22.2 | (21.3–23.2) | 22.5 | (21.5–23.6) | 11.9 | (6.4–21.1) | |
| Grade 10 | 22.6 | (21.7–23.5) | 22.5 | (21.5–23.5) | 22.1 | (12.7–35.7) | |
| Grade 11 | 23.5 | (22.7–24.2) | 23.3 | (22.5–24.2) | 33.6 | (21.0–49.1) | |
| Grade 12 | 31.7 | (30.3–33.2) | 31.7 | (30.2–33.2) | 32.4 | (20.7–46.8) | |
| **Lifetime TBI** | | | | | | | *** |
| Yes | 20.8 | (19.6–22.1) | 19.5 | (18.0–21.0) | 41.5 | (27.1–57.5) | |
| No | 79.2 | (77.9–80.4) | 80.5 | (79.0–82.0) | 58.5 | (42.5–72.9) | |
| **Hazardous Drinking** | | | | | | | *** |
| AUDIT = > 8 | 21.1 | (19.6–22.7) | 20.6 | (19.1–22.1) | 57.5 | (45.3–68.9) | |
| AUDIT < 8 | 78.9 | (77.3–80.4) | 79.4 | (77.9–80.9) | 42.5 | (31.2–54.7) | |
| **Suicidal thoughts / attempts** | | | | | | | *** |
| Yes | 13.2 | (12.2–14.2) | 12.9 | (12.0–13.9) | 33.1 | (21.6–46.9) | |
| No | 86.8 | (85.8–87.8) | 87.1 | (86.1–88.0) | 66.9 | (53.1–78.4) | |

Note: All percentages, point estimates and 95% confidence intervals reflect weighted estimates correcting for the sampling design

* p<0.05

** p<0.01

*** p<0.001.

**Table 2. Logistic regression model of self-reported problem gambling by demographic characteristics, lifetime traumatic brain injuries, hazardous drinking and suicidal thoughts or attempts among secondary school students, OSDUHS 2011–2015, Ontario, Canada (N = 9,198).**

|  | Model 1 | | | Model 2 | | | Model 3 | | |
|---|---|---|---|---|---|---|---|---|---|
|  | AOR | 95% CI | p | AOR | 95% CI | P | AOR | 95% CI | p |
| **Male** | 3.68 | 1.68 | 8.07 | *** | 4.94 | 2.31 | 10.55 | *** | 4.73 | 2.18 | 10.23 | *** |
| **Grade** | 1.16 | 0.95 | 1.42 | NS | 0.95 | 0.74 | 1.22 | NS | 0.97 | 0.75 | 1.24 | NS |
| **Lifetime TBI** | 2.80 | 1.43 | 5.49 | *** |  |  |  |  | 2.02 | 1.06 | 3.88 | * |
| **Hazardous Drinking** |  |  |  |  | 5.04 | 2.79 | 9.12 | *** | 4.48 | 2.56 | 7.83 | *** |
| **Suicidal thoughts / attempts** |  |  |  |  | 3.27 | 1.75 | 6.11 | *** | 3.11 | 1.66 | 5.84 | *** |

In model 3, the effect of TBI was still significant (AOR 2.02; 95% CI:1.06–3.88), although the odds ratio was reduced from Model 1. The association of both hazardous drinking (AOR 4.48; 95% CI: 2.56–7.83) and suicidality (AOR 3.11; 95% CI: 1.66–5.84) with problem gambling was also reduced slightly. In addition, we tested for interaction effects between TBI and suicidality and TBI and hazardous drinking. Neither interaction was statistically significant (results available upon request).

## Discussion

The primary objective of this study was to examine the association between TBI and problem gambling among a population-based sample of secondary school students from Ontario, Canada. Consistent with recent research among adults [17], we found that students with a history of TBI had 2 times the odds of a gambling problem compared to those without TBI, after controlling for hazardous drinking and suicidality. These findings provide additional support for an association between TBI and gambling problems, and extend this observation to the adolescent population [26–28, 51].

While the unadjusted odds ratio indicated a strong relationship between problem gambling and lifetime TBI, this relationship may have been due to common underlying covariates. Controlling for hazardous drinking was therefore important because hazardous drinking has been previously shown to be related to TBI [21, 22, 39] and to problem gambling [2, 52, 53]. As well, controlling for suicidality was relevant because previous research has also shown that suicidal ideation and behavior were linked with TBI [22, 54] and with excessive/pathological or problematic gambling [14, 55, 56]. The results suggest that there is an association between TBI and problem gambling, which is not fully explained by hazardous drinking or suicidality. The lack of interactions also suggests that the association does not vary by one of these confounding variables.

We also observed that adolescent problem gambling was significantly associated with increased suicidality, which is consistent with previous studies with adolescent samples [14, 56, 57]. Similarly consistent with previous research, adolescent problem gambling was also significantly associated with hazardous or harmful drinking [11–13, 15, 58, 59]. These findings highlight the importance, for prevention and intervention efforts, of recognizing the comorbidity of problem gambling with other mental health and substance problems in adolescent populations. They also support the validity of theoretical models that highlight the clustering of gambling and other problem behaviors in adolescence [60, 61].

There may be several possible explanations for the association between problem gambling and TBI including injury to particular parts of the brain such as the frontal lobes [62], underlying biological causes, or shared personality factors [17, 63–65]. It could also be that dispute over gambling outcomes has led to the injury. More research is needed to examine these

mechanisms, and longitudinal studies are needed to disentangle the temporal sequence of the association between TBI and problem gambling.

Recognition of this association has important implications for prevention and treatment [62]. In particular, given the devastating financial and interpersonal consequences that can result from excessive gambling, physicians and other health care professionals who treat adolescent patients with TBI should consider discussing problem gambling with their patients and caregivers as a possible consequence of TBI. TBI rehabilitation should include attention to why individuals cannot win at gambling and how to stay in control emotionally to help to avoid experiencing the severe financial consequences of problem gambling. This is particularly important given that TBI may impair attention, memory, and social functioning. Similarly, problem gamblers should be asked about past TBI because the presence of such injuries may suggest that the gambling treatment process needs to take into account the neuro-cognitive and social consequences of the TBI. The association between TBI and problem gambling may also help to explain why some problem gamblers have difficulties in responding to treatment due to previously un-recognized neuro-cognitive and social deficits caused by prior TBI.

The results of the current study have important implications for understanding the link between traumatic brain injury and problem gambling. TBI is associated with a large range of mental health and addiction problems [22–24, 34, 66–68], which highlights the importance of prevention and management efforts targeting psychiatric risks and outcomes related to TBI [34]. According to Bryant et al. [24] about one-third of TBI survivors experience significant mental health service needs as a result of depression, anxiety, and substance abuse. Public health initiatives are needed to address the huge mental health burden associated with these injuries [24]. New approaches are needed to identify emergent psychiatric disorders following TBI so that early interventions can be applied and facilitate recovery [24]. The current study suggests that problem gambling is an additional risk for this group. Mental health professionals and others can play a critical role in system-level changes that will improve the treatment and health outcomes of TBI in childhood and adolescence. Specifically, progress can be made through education to individuals and to increased public awareness of the effects of TBI on mental health. Advocacy for those who experience a TBI, and working towards more comprehensive clinical practice that takes into account the wide range of effects of TBI on mental health including substance abuse and problem gambling are other important policy implications of our study and other studies [24, 34]. Chan et al. [33] argue for the need to integrate care linking treatment for TBI with treatment of mental health and addictions which are currently not well coordinated. We also posit that the significant associations we have identified support the need for sustained public health interventions to prevent the comorbid mental health, TBI and addiction behaviors that are associated with problem gambling among adolescents.

## Limitations

This study is a cross-sectional study, and therefore it is not possible to determine the causal relationships involved in the observed relationships. As noted, previous studies have reported that compared to the general population, increased aggressiveness, impulsivity, risk-taking, and poor mental health are associated with both gambling problems [64, 69–71] and TBI [21, 65, 72–76]. In terms of causal pathways it could be that (1) TBI leads to gambling problems or vice versa, (2) both TBI and problem gambling stem from an underlying characteristic such as impulsiveness, (3) the link between TBI and problem gambling is mediated by risk-taking, hazardous drinking, suicidality, impulsivity, poor mental health, or other factors. Longitudinal research is needed to clarify the nature of the causal pathways involved.

Another limitation to the present study is that the OSDUHS did not obtain detailed information on the severity of the injury, the part of the brain injured, or the temporal sequence of the onset of problem gambling and the head injury. In addition, we did not include milder forms of TBI. Also, because the data were based on self-report, the results may be subject to bias resulting from self-reports such as potential errors in recalling details of injury. Finally, the sample only includes non-institutionalized youth attending publicly funded schools which might miss some of the more severe cases of TBI.

## Conclusions

In this study we replicated previous research on the link between TBI and problem gambling among adults [17], and extended the findings to an adolescent population. To our knowledge, this is the first study to specifically focus on problem gambling and TBI in adolescence. Further, this study demonstrates that the association between problem gambling and TBI is not simply attributable to the comorbidity of gambling and risky alcohol use or suicidality. The strong link between suicide ideation and problem gambling also replicates previous research [14]. This link between injury to the brain and problem gambling may also advance our understanding of why some people are unable to control their gambling. These findings suggest that it is important to target problem gambling prevention initiatives towards adolescents who have experienced a TBI, and for those treating problem gamblers to consider the possible effects of prior TBI in developing rehabilitation and treatment plans.

## Supporting information

**S1 Table. Alternative logistic regression model of self-reported problem gambling by demographic characteristics, lifetime traumatic brain injuries, hazardous drinking and suicide attempts among secondary school students, OSDUHS 2011–2015, Ontario, Canada ($N$ = 9,198).**
(DOCX)

## Acknowledgments

The ideas expressed are those of the authors and do not necessarily reflect those of the Centre for Addiction and Mental Health, the University of Toronto, or the funder of this research project. The authors acknowledge the Institute for Social Research at York University for administering the data collection.

## Author Contributions

**Conceptualization:** Nigel E. Turner, Steven Cook, Jing Shi.

**Formal analysis:** Steven Cook.

**Funding acquisition:** Nigel E. Turner, Robert E. Mann.

**Methodology:** Tara Elton-Marshall.

**Project administration:** Hayley Hamilton.

**Supervision:** Robert E. Mann.

**Validation:** Gabriela Ilie, Nico Trajtenberg, Michael D. Cusimano.

**Writing – original draft:** Nigel E. Turner, Steven Cook.

**Writing – review & editing:** Nigel E. Turner, Steven Cook, Jing Shi, Tara Elton-Marshall, Hayley Hamilton, Gabriela Ilie, Christine M. Wickens, André J. McDonald, Nico Trajtenberg, Michael D. Cusimano, Robert E. Mann.

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
