## [Decision Letter · Decision Letter 0]

15 Jul 2020

PONE-D-20-11819

Traumatic Brain Injuries and Problem Gambling: Evidence from a population-based study of secondary students in Ontario, Canada.

PLOS ONE

Dear Dr. Turner,

Thank you for submitting your manuscript to PLOS ONE. After careful consideration, we feel that it has merit but does not fully meet PLOS ONE’s publication criteria as it currently stands. Therefore, we invite you to submit a revised version of the manuscript that addresses the points raised during the review process.

I apologise for the delay in your manuscript. Unfortunately reviewers had agreed and then not completed the reviews. Rather than further impede the review process, I have decided to return the manuscript for you with the review below. This detailed review captures key adjustments to the manuscript. It may be that the resubmission is sent to an additional reviewer.

We look forward to receiving your revised manuscript.

Kind regards,

Simone Rodda

Academic Editor

PLOS ONE

Journal Requirements:

2. You indicated that you had ethical approval for your study. In your Methods section, please ensure you have also stated whether you obtained consent from parents or guardians of the minors included in the study or whether the research ethics committee or IRB specifically waived the need for their consent. Please also specify: 1) whether the ethics committee approved the verbal consent procedure, 2) why written consent could not be obtained, and 3) how verbal consent was recorded. If the need for written consent or parental consent was waived by the ethics committee, please include this information.

3.We note that you have indicated that data from this study are available upon request. PLOS only allows data to be available upon request if there are legal or ethical restrictions on sharing data publicly. For information on unacceptable data access restrictions, please see http://journals.plos.org/plosone/s/data-availability#loc-unacceptable-data-access-restrictions.

4.Thank you for stating the following in the Competing Interests section:

[Turner has received funding from the Ontario Ministry of Health and Long Term Care (OMHLTC), Gambling Research Exchance (GREO) and from Ontario Lottery and Gaming (OLG). In all cases the contract included guarantees of independence and intellectual property rights for the researcher. Turner has also acted as a consultant on gambling problems for various government and legal entities.

The rest of the authors report no conflicts of interest, however the following authors  reported funding from various sources.

Jing Shi has received funding from GREO

Michael Cusimano reports funding from CIHR, CIMVHR, IBM, Mitacs.

Chrisinte Wilkensons reports funding from CIHR, OMHLTC, Higher Education Quality Council of Ontario, Workplace Safety and Insurance Board, Alcohol Countermeasures Systems Corp., and the CAMH Foundation (Caskey-Francis Family Award).

Robert Mann has received funding from CIHR, OMHLTC, and Ontario Ministry of Transportation,].

5. Your ethics statement must appear in the Methods section of your manuscript. If your ethics statement is written in any section besides the Methods, please move it to the Methods section and delete it from any other section. Please also ensure that your ethics statement is included in your manuscript, as the ethics section of your online submission will not be published alongside your manuscript.

Reviewers' comments:

Reviewer's Responses to Questions

**Comments to the Author**

1. Is the manuscript technically sound, and do the data support the conclusions?

Reviewer #1: Yes

2. Has the statistical analysis been performed appropriately and rigorously? 

Reviewer #1: Yes

3. Have the authors made all data underlying the findings in their manuscript fully available?

Reviewer #1: Yes

4. Is the manuscript presented in an intelligible fashion and written in standard English?

Reviewer #1: No

5. Review Comments to the Author

Reviewer #1: The authors conducted an interesting research aimed at evaluating the relation between traumatic brain injury history and gambling severity in adolescents. Below, find some major and minor points that need to be addressed in order to improve the manuscript. After these changes, it will be possible to judge whether the research is suitable for publication.

Major revisions

In the first lines of the Introduction, the authors report the prevalence rates of gambling in adults and adolescents. About that, I suggest some revisions. First, the articles cited to report the prevalence rates are too old and should be updated (for example, Korn et al., 1999; Gupta et al., 1998; Turner et al., 2008, 2008, 2011). Second, in the line 71-77, the authors report adolescent gambling, then adult gambling, and then adolescent gambling again. I would suggest two options: 1) talk about adult and then adolescent gambling; 2) focus the introduction on adolescent gambling exclusively.

In the Introduction, line 86-89, the authors report studies that linked substance abuse, anxiety, and depression to problem gambling, but in adults and not in adolescents. I think the authors should mention the literature on adolescent gambling, given that their study is about adolescent gambling.

In the Introduction, line 96, the statement “Given the relationships of TBI with alcohol consumption and other substance use and mental health problems, it seems plausible that adolescent TBI may also be related to problem gambling.” Lacks of reference.

The rationale of the study is poor. There are no information of the process by which brain injury causes impulsive behaviors. There are little data about adolescent gambling. I think that the Introduction is not very persuasive and need to be improved.

It is not enough to refer the reader to other articles to know the procedure of the study (“Further information on the OSDUHS, including the sampling procedures and weighting of data, is available in references”). I believe it is necessary to report the procedure.

The first question about suicide attempt “did you ever seriously consider attempting suicide” is a little tricky. It is well known that suicide thoughts are common among people, especially in adolescents but that a thought not always is followed by an action. I know what you intend with “seriously” but I have doubts about the way in which adolescents could have interpreted this question, so I would recommend to use only the second question.

Discussion. The authors state “While there may be several possible explanations for this association, including underlying biological causes and shared personality factors (35, 54-56) recognition of this relationship has important implications for prevention and treatment.” What implications do they intend? It is necessary to specify.

Discussion, line 247. “We also observed that adolescent problem gambling was significantly associated with 248 increased suicidality, which is consistent with previous studies (11, 57, 58).”

The studies you cited are on adults or adolescents?

Discussion, line 248. “Similarly consistent with previous research, adolescent problem gambling was also significantly associated with hazardous or harmful drinking (4, 59, 60)”. The studies cited are too old if you consider that the association between alcohol use and gambling in adolescents has been highly investigated (see for example “Decision-Making, Cognitive Distortions and Alcohol Use in Adolescent Problem and Non-problem Gamblers: An Experimental Study”, “The Associations Between Maladaptive Personality Traits, Craving, Alcohol Use, and Adolescent Problem Gambling: An Italian Survey Study”, “Adolescent alcohol-drinking frequency and problem-gambling severity: Adolescent perceptions regarding problem-gambling prevention and parental/adult behaviors and attitudes”, or “A review of gambling disorder and substance use

disorders”).

The discussion is too poor and need to be enriched. For example, with clinical implications. Moreover, it is not clear to me what novelty results this study provides to the literature.

In general the references are too old and should be updated, especially the literature about adolescent gambling.

Minor revisions

The paper needs a more careful reading by the authors, as I found some typo, such as:

- In the Introduction, line 75, it should be “affects” and not “affect”.

- Introduction, line 105, “Similarly, recent a case-control study by Bhatti et al”.

- Introduction, line 109, “no studies have yet”.

- Methods, line 129. The authors say “The total sample size was 9,198 secondary school students 129 (52.0%)”. 52% of what?

- Measures, line 151. The “Diagnostic and Statistical Manual of Mental Disorders, 5th edition (45)” has another font and colour as compared with the paper.

- Measure, line 156. Given that it is also mentioned later, “which was developed by the World Health Organization (47)” should be removed. In the same way, the construct measured by the AUDIT and its number of items are reported twice.

References should be checked. For example, Derevensky J, Gupta R. Adolescents with gambling problems: A synopsis of our current 459 knowledge. Journal of Gambling Issues. 2004;10 lacks of number of pages.

6. PLOS authors have the option to publish the peer review history of their article (what does this mean?). If published, this will include your full peer review and any attached files.

Reviewer #1: No

---

## [Author Response · Author response to Decision Letter 0]

14 Aug 2020

Response to Editorial and Reviewer comments.

Comment: 1. Please ensure that your manuscript meets PLOS ONE's style requirements, 

Response: Thank you. We have ensured that the manuscript meets style requirements.

Comment: 2. You indicated that you had ethical approval for your study. In your Methods section, please ensure you have also stated whether you obtained consent from parents or guardians of the minors included in the study or whether the research ethics committee or IRB specifically waived the need for their consent. Please also specify: 1) whether the ethics committee approved the verbal consent procedure, 2) why written consent could not be obtained, and 3) how verbal consent was recorded. If the need for written consent or parental consent was waived by the ethics committee, please include this information.

Response: We have included this statement in the methods section: “An active consent method was used. A description of the study and a consent form was sent home with the students. To participate in the survey, written parental consent and written student assent were required. " 

Comment: 3.We note that you have indicated that data from this study are available upon request. PLOS only allows data to be available upon request if there are legal or ethical restrictions on sharing data publicly. For information on unacceptable data access restrictions, please see http://journals.plos.org/plosone/s/data-availability#loc-unacceptable-data-access-restrictions.

Response: The data restrictions have no relationship to “personal interests, such as patents or potential future publications”, and is not due to “analysis of proprietary data.” CAMH requires external researchers to complete an application for access to the data and upon approval a data sharing agreement is required. Due to institutional restrictions data cannot be made available in the manuscript, the supplemental files or a public repository. Data are available to all researchers after an institutional application is approved. To access data contact osduhs@camh.ca for further instructions. Full reports of the public data files and the surveys administered can be accessed online: www.camh.ca/osduhs

Comment: a) If there are ethical or legal restrictions on sharing a de-identified data set, please explain them in detail (e.g., data contain potentially identifying or sensitive patient information) and who has imposed them (e.g., an ethics committee). Please also provide contact information for a data access committee, ethics committee, or other institutional body to which data requests may be sent.

Response: Due to institutional restrictions data cannot be made available in the manuscript, the supplemental files or a public repository. Data are available to all researchers after an institutional application is approved and a data sharing agreement is established. To access data contact osduhs@camh.ca for further instructions. Full reports of the public data files and the surveys administered can be accessed online: www.camh.ca/osduhs

Comment: 4.Thank you for stating the following in the Competing Interests section:

[see above in cover letter]

Please confirm that this does not alter your adherence to all PLOS ONE policies on sharing data and materials, by including the following statement: "This does not alter our adherence to PLOS ONE policies on sharing data and materials.” 

Response: We confirm that neither our funding sources or competing interests alter our adherence to PLOS ONE polices on sharing data and material. The access issue has to do with our ethics board. We have added the statement as request.

Comment: 5. Your ethics statement must appear in the Methods section of your manuscript. If your ethics statement is written in any section besides the Methods, please move it to the Methods section and delete it from any other section. Please also ensure that your ethics statement is included in your manuscript, as the ethics section of your online submission will not be published alongside your manuscript.

Response: Thank you, We’ve moved the ethics description to the first section of the Methods section (Sample).

Comment: 4. Is the manuscript presented in an intelligible fashion and written in standard English?

Response: The manuscript has been carefully copy-edited. 

Comment: In the first lines of the Introduction, the authors report the prevalence rates of gambling in adults and adolescents. About that, I suggest some revisions. First, the articles cited to report the prevalence rates are too old and should be updated (for example, Korn et al., 1999; Gupta et al., 1998; Turner et al., 2008, 2008, 2011). 

Response: We have updated the references as requested.

Comment: Second, in the line 71-77, the authors report adolescent gambling, then adult gambling, and then adolescent gambling again. I would suggest two options: 1) talk about adult and then adolescent gambling; 2) focus the introduction on adolescent gambling exclusively.

Response: We have removed most references to adult gambling and the focus is now on adolescent gambling. 

Comment: In the Introduction, line 86-89, the authors report studies that linked substance abuse, anxiety, and depression to problem gambling, but in adults and not in adolescents. I think the authors should mention the literature on adolescent gambling, given that their study is about adolescent gambling.

In the Introduction, line 96, the statement “Given the relationships of TBI with alcohol consumption and other substance use and mental health problems, it seems plausible that adolescent TBI may also be related to problem gambling.” Lacks of reference.

Response: We have updated and added references as requested.

Comment: The rationale of the study is poor. There are no information of the process by which brain injury causes impulsive behaviors. There are little data about adolescent gambling. I think that the Introduction is not very persuasive and need to be improved.

Response: We appreciate the reviewer’s interest in a stronger rationale. Research on brain injury and adolescent gambling is currently in its infancy, and we have revised the introduction to strengthen the rationale. Previous research has demonstrated links between traumatic brain injury (TBI) and risk behaviours. This was the reason why we decided to examine the potential association with gambling. However, this is the first study that we know of to document the association between TBI and gambling in a youth population. At this point, we do not know why brain damage is associated with problem gambling. We speculate that those people who have damage to their frontal lobe become problem gamblers based on previous studies demonstrating the relation between brain injuries and other risk behaviour. It could also be that gambling contributed to TBI (e.g., fights that result from a game dispute). Additional research examining these mechanisms is clearly needed, and as noted we have revised the Introduction and Discussion to highlight these points.

Comment: It is not enough to refer the reader to other articles to know the procedure of the study (“Further information on the OSDUHS, including the sampling procedures and weighting of data, is available in references”). I believe it is necessary to report the procedure.

Response: We have added additional details about sampling procedures and survey weights. 

Comment: The first question about suicide attempt “did you ever seriously consider attempting suicide” is a little tricky. It is well known that suicide thoughts are common among people, especially in adolescents but that a thought not always is followed by an action. I know what you intend with “seriously” but I have doubts about the way in which adolescents could have interpreted this question, so I would recommend to use only the second question.

Response: We acknowledge, as suggested by the reviewer, that the rate of suicide ideation is higher than suicide attempts. Even so, in our sample suicide ideation itself is not very common and was only reported by 13% of the sample. In addition, we believe suicide ideation is important for us to include in our analyses because it captures mental health problems severe enough that these youth are thinking about taking their life and is thus an important marker of psychological distress even if it does not necessarily result in suicide. We feel that suicide ideation is an important construct, and taps into a growing body of literature that has demonstrated the association between psychological distress and problem gambling among adolescents (c.f., Cook et al., 2015). 

That said, we recognize the importance of the comment and therefore repeated the analysis with ‘suicide attempt’ only. We believe this type of ‘sensitivity analysis’ is important, and we wanted to ensure that our focal variable did not change when we included the different measure of suicide. For transparency, we have included the results in the table below with suicide attempt instead of suicide ideation. We have also included this alternative Table in supplementary materials. As you can see, the AOR of suicide attempt is (predictably) higher (and has wider confidence intervals, reflecting small cell counts), but this measurement did not change the AOR of the other variables or their meaning. Because we are interested in ideation itself is an indication of emotional distress, which ties directly into the extant research, we decided to keep our measure of suicide ideation in the revised manuscript and hope the reviewer understands this decision. 

Table S1: Alternative logistic regression model of self-reported problem gambling by demographic characteristics, lifetime traumatic brain injuries, hazardous drinking and suicide attempts among secondary school students, OSDUHS 2011-2015, Ontario, Canada (N=9,198).

 Model 1 Model 2 Model 3

 AOR 95% CI p AOR 95% CI p AOR 95% CI p

Male 3.68 1.68 8.07 *** 5.08 2.46 10.46 *** 4.91 2.34 10.30 ***

Grade 1.16 0.95 1.42 NS 0.95 0.74 1.22 NS 0.96 0.75 1.23 NS

Lifetime TBI 2.80 1.43 5.49 *** 1.94 1.02 3.70 *

Hazardous Drinking 5.09 2.83 9.17 *** 4.58 2.54 7.93 ***

Suicide attempts 6.41 2.52 16.27 *** 5.71 2.18 14.91 ***

Comment: Discussion. The authors state “While there may be several possible explanations for this association, including underlying biological causes and shared personality factors (35, 54-56) recognition of this relationship has important implications for prevention and treatment.” What implications do they intend? It is necessary to specify.

Response: Thank you for the comment. We have moved this line toward the end of the Discussion and we have expanded upon the potential implications of these findings. 

Comment: Discussion, line 247. “We also observed that adolescent problem gambling was significantly associated with increased suicidality, which is consistent with previous studies (11, 57, 58).”

The studies you cited are on adults or adolescents?

Response: all three of these studies were on adolescents. For example in Cook et al. (2015), we used logistic regression to determine the relationship in adolescent data between problem gambling and suicide attempts as well as delinquent behaviours. The studies by Nower (2004) and Derevensky (2004) are also focused on youth. In addition, other studies have also reported suicidal ideation amongst problem gambler adolescents and adults. I’ve added an additional citation from 2020 by Farhat to this reference list and removed Derevensky’s 2004 citation. We have also clarified in the manuscript that these studies involved adolescent samples. 

Comment: Discussion, line 248. “Similarly consistent with previous research, adolescent problem gambling was also significantly associated with hazardous or harmful drinking (4, 59, 60)”. The studies cited are too old if you consider that the association between alcohol use and gambling in adolescents has been highly investigated (see for example “Decision-Making, Cognitive Distortions and Alcohol Use in Adolescent Problem and Non-problem Gamblers: An Experimental Study”, “The Associations Between Maladaptive Personality Traits, Craving, Alcohol Use, and Adolescent Problem Gambling: An Italian Survey Study”, “Adolescent alcohol-drinking frequency and problem-gambling severity: Adolescent perceptions regarding problem-gambling prevention and parental/adult behaviors and attitudes”, or “A review of gambling disorder and substance use

disorders”).

Response: Thank you. We have updated the citations and removed the older references.

Comment: The discussion is too poor and need to be enriched. For example, with clinical implications. Moreover, it is not clear to me what novelty results this study provides to the literature.

Response: We appreciate the reviewer’s interest in a more extensive Discussion and have revised the manuscript to provide this. This is the first study to demonstrate that traumatic brain injury is associated with problem gambling in adolescents. As far as we know, this is the first study to specifically focus on problem gambling and TBI in adolescence. In addition we have greatly expanded that part of the discussion

Comment: In general the references are too old and should be updated, especially the literature about adolescent gambling.

Response: We have updated the paper with more recent citations.

Comment: The paper needs a more careful reading by the authors, as I found some typo, such as:

- In the Introduction, line 75, it should be “affects” and not “affect”.

- Introduction, line 105, “Similarly, recent a case-control study by Bhatti et al”.

- Introduction, line 109, “no studies have yet”.

- Methods, line 129. The authors say “The total sample size was 9,198 secondary school students 129 (52.0%)”. 52% of what?

- Measures, line 151. The “Diagnostic and Statistical Manual of Mental Disorders, 5th edition (45)” has another font and colour as compared with the paper.

Response: Thank you we have made these corrections and further proofed the paper. Note that it should have read been 52% females.

Comment: - Measure, line 156. Given that it is also mentioned later, “which was developed by the World Health Organization (47)” should be removed. In the same way, the construct measured by the AUDIT and its number of items are reported twice.

Response: Thank you for pointing out this duplication, and we have made this correction

Comment: References should be checked. For example, Derevensky J, Gupta R. Adolescents with gambling problems: A synopsis of our current 459 knowledge. Journal of Gambling Issues. 2004;10 lacks of number of pages.

Response: the journal is an online journal and only has an HTML format for this particular paper, and it doesn’t have pages per se, however it was printed out to 12 pages, so I’ve added in as the 1-12 as page numbers.

---

## [Editor Report · Decision Letter 1]

11 Sep 2020

Traumatic Brain Injuries and Problem Gambling in Youth: Evidence from a population-based study of secondary students in Ontario, Canada.

PONE-D-20-11819R1

Dear Dr. Turner,

We’re pleased to inform you that your manuscript has been judged scientifically suitable for publication and will be formally accepted for publication once it meets all outstanding technical requirements.

Kind regards,

Simone Rodda

Academic Editor

PLOS ONE

Additional Editor Comments (optional):

Thank you to the authors for responding to the extensive comments.

---

## [Editor Report · Acceptance letter]

22 Sep 2020

PONE-D-20-11819R1 

Traumatic Brain Injuries and Problem Gambling in Youth: Evidence from a population-based study of secondary students in Ontario, Canada. 

Dear Dr. Turner:

I'm pleased to inform you that your manuscript has been deemed suitable for publication in PLOS ONE. Congratulations! Your manuscript is now with our production department. 

Kind regards, 

on behalf of

Dr. Simone Rodda 

Academic Editor

PLOS ONE